# Extinct type of human parvovirus B19 persists in tonsillar B cells

Lari Pyöriä[1,*], Mari Toppinen[1,*], Elina Mäntylä[2], Lea Hedman[1,3], Leena-Maija Aaltonen[4], Maija Vihinen-Ranta[2], Taru Ilmarinen[4], Maria Söderlund-Venermo[1], Klaus Hedman[1,3] & Maria F. Perdomo[1]

Parvovirus B19 (B19V) DNA persists lifelong in human tissues, but the cell type harbouring it remains unclear. We here explore B19V DNA distribution in B, T and monocyte cell lineages of recently excised tonsillar tissues from 77 individuals with an age range of 2–69 years. We show that B19V DNA is most frequent and abundant among B cells, and within them we find a B19V genotype that vanished from circulation >40 years ago. Since re-infection or re-activation are unlikely with this virus type, this finding supports the maintenance of pathogen-specific humoral immune responses as a consequence of B-cell long-term survival rather than continuous replenishment of the memory pool. Moreover, we demonstrate the mechanism of B19V internalization to be antibody dependent in two B-cell lines as well as in *ex vivo* isolated tonsillar B cells. This study provides direct evidence for a cell type accountable for B19V DNA tissue persistence.

[1] Department of Virology, University of Helsinki, Helsinki 00290, Finland. [2] Department of Biological and Environmental Science, and Nanoscience Center, University of Jyväskylä, Jyväskylä 40500, Finland. [3] Helsinki University Hospital, Helsinki 00290, Finland. [4] Department of Otorhinolaryngology-Head and Neck Surgery, Helsinki University Hospital, Helsinki 00290, Finland. * These authors contributed equally to this work. Correspondence and requests for materials should be addressed to M.F.P. (email: maria.perdomo@helsinki.fi).

Parvovirus B19 (B19V) infection affects most often children, with exposure rates generally over 50% by adulthood[1]. The virus circulates worldwide, with current infections mainly due to genotype 1 (ref. 2). Of the other two variants that are known, genotype 2 disappeared from circulation around 1970 (refs 3,4) and genotype 3 has been described to circulate endemically in some regions such as Ghana, Brasil and India[5–8].

After primary infection, B19V DNA persists lifelong in several human tissues such as tonsils, testicles, kidneys, muscle, salivary glands, thyroid, skin, liver, heart, brain, bone marrow and bone[3,4,9–11]. However, nothing is known on the specific cell type(s) that harbours it throughout time.

B19V replicates in erythroid progenitor cells of the bone marrow with primary infection occurring via the globoside receptor and the α5β1 integrin and Ku80 co-receptors[12–14] but uptake has also been shown to occur through antibody-dependent enhancement (ADE) in monocytes[15] and endothelial cells[16]. The short lifetime of these cells, however, does argue against them being the host of this virus' DNA for years after primary infection. Instead, an appealing alternative may be granted by the memory cells that reside in lymphoid organs since their lifespan has been estimated to exceed decades based on the length of immune protection after infection or vaccination[17].

Hence, in the present study, we evaluate the distribution of B19V DNA in lymphoid cells of recently excised tonsillar tissues. Moreover, we analyse the virus type present, having previously shown[11] that the B19V genotype 2 is a reliable indicator of the age of a tissue. We found the B19V DNA to be primarily distributed in B cells and most importantly, we detected in four adults the extinct genotype 2, thus providing further evidence of this cell type as long-term reservoir of B19V DNA. This finding also enacts as a suitable marker of the longevity of these cells. Moreover, we show ADE to be a mechanism for B19V uptake into B cells in vitro.

## Results

**B19V DNA persistence in tonsils**. B19V DNA has been previously detected in human tonsils[3] but the specific cell types where it persists lifelong are still unknown. We analysed the B19V-DNA distribution in enriched lymphoid cell suspensions of recently excised tonsillar tissues from 77 individuals who underwent tonsillectomy at the Department of Otorhinolaryngology-Head and Neck Surgery of Helsinki University Hospital. Previous findings by Medina et al.[18] show that mechanical homogenization alone is not sufficient to release long-lived cells that are associated to connective tissue-rich areas of the tonsil, such as resident plasma cells. Hence a two-step process of tissue homogenization involving mechanical disaggregation followed by collagenase digestion of the residual tonsillar tissue was developed. Each of the resulting cell suspensions was analysed independently for the presence of B19V DNA by an in-house Pan-B19V qPCR amplifying the NS1 region, and the viral copy numbers were normalized to cell counts by quantification of the single copy RNase P gene.

B19V DNA was detected in 26% (20/77) of the total cell populations obtained by mechanical homogenization alone as opposed to 43% (33/77) in those cells released by subsequent collagenase digestion. Moreover, in the latter, the median B19V-DNA copy numbers were 18-fold higher ($P < 0.001$, Mann–Whitney $U$ asymptotic sig. (two-sided test; Fig. 1a)).

The B, T and monocyte/macrophage (M) cells were enriched from each tonsillar preparation by positive selection with magnetic beads. The cell fraction purities were: B 96.8 ± 0.9%, T 95.4 ± 1.2%, M 93.9 ± 1.9% (mean ± s.d. of 6 replicates).

B19V DNA was preferentially distributed in the B cells of the collagenase-treated preparations (33/33 individuals) which contained also the highest viral loads: median 6.91E1 copies/1E6 cells (95% confidence interval (CI): 2.26E1–9.53E1 B19V-DNA copies /1E6 cells) as compared to $1.7E - 1$ copies/1E6 cells (95% CI: 0.00–3.08) in the fraction resulting from homogenization alone (Fig. 1c). The difference was statistically significant ($P < 0.001$, Mann–Whitney $U$ asymptotic sig. (two-sided test)). The B19V-DNA positivity of the B-cell fractions from collagenase-treated tissues was confirmed with a second B19V qPCR amplifying a distinct region (VP1 gene) of the viral genome. There was a strict correlation between both qPCRs, with similar copy numbers (Supplementary Fig. 1).

The Pan-B19V qPCR products of the B cells released with collagenase were sequenced to determine the persisting B19V genotype. Strikingly, among the six B19V genopositive adults older than 45 years of age (45 to 69; mean 55), four had in their B cells the extinct genotype 2 (median 1.01E2 copies /1E6 cells). All other individuals ($n = 29$) had genotype 1 (Supplementary Table 1).

Interestingly, two children of 6 and 8 years had at least one log higher B19V DNA loads than all the other individuals (3.66E3 and 9.77E3 copies/1E6 cells, respectively). In these two subjects, the highest amounts of B19V DNA were detected in the M population of the homogenized cell suspension (2.50E4 and 5.38E4 copies /1E6 cells) albeit B19V-DNA levels were high in other cell fractions as well (Supplementary Fig. 2). These two children were IgM negative and non-viremic.

Serum samples available from 74/77 individuals were tested for B19V-IgG and -IgM antibodies. Altogether, 42% (31/74) were B19V-IgG positive and none were B19V-IgM positive. The B19V-DNA prevalence in tissue and the IgG serostatus of the individuals showed absolute correlation (Supplementary Fig. 3).

**Epstein-Barr virus DNA persistence in tonsils**. The prevalence of B19V DNA was compared to that of EBV because of the latter's known persistence in B cells[19].

The EBV-DNA prevalence was 73% (24/33) and 70% in the B19V DNA-positive and -negative tonsils, respectively, thus lacking significance ($P = 0.827$, Pearson's $\chi^2$-test).

EBV DNA was found preferentially distributed in the B cells, with a median of 2.10E2 and 1.99E2 copies/1E6 cells among the total cell suspensions obtained with or without collagenase treatment, respectively, in the B19V DNA-positive tonsillar preparations (Fig. 1b,d).

**B19V- and EBV-DNA persistence in circulating lymphoid cells**. The prevalences of B19V and EBV DNA among circulating B cells were evaluated in peripheral blood mononuclear cells (PBMCs) as well as in CD19+ cells isolated from seven B19V-seropositive and one B19V-seronegative, asymptomatic staff members. Both the PBMCs and B cells from all the subjects were PCR-negative for B19V DNA, but positive for EBV DNA in 3/8 individuals (mean 7.60E1 and 4.81E2 copies / 1E6 cells in PBMC and B cells, respectively; Supplementary Table 2).

Compared to tonsillar B cells, the circulating CD19+ cells were globoside negative by flow cytometry (Supplementary Fig. 4d).

**Characterization of tonsillar B cells**. The distribution of B19V DNA among naive and memory B cells was studied from collagenase-treated tonsillar cell suspensions of 12 individuals with the highest copy numbers (median 1.06E2 copies/1E6 cells). The cells were stained with anti-CD19 antibodies, to gate the B-cell population, together with CD27 (mature) and IgD (naive) antibodies. Four subpopulations were sorted: CD27+

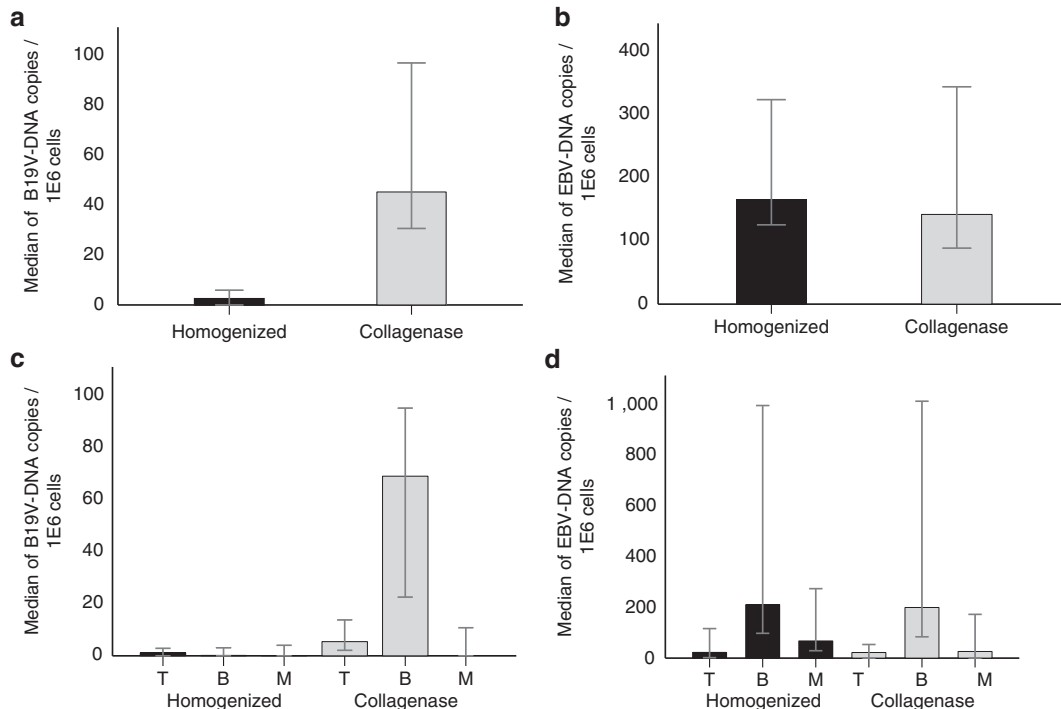

**Figure 1 | Viral DNA copies in tonsillar tissue.** B19V- and EBV-DNA copies were measured by qPCR and normalized to cell numbers with the human single-copy gene *RNase P*. Top panels: median copy numbers of B19V DNA (**a**) and EBV DNA (**b**) in tonsillar cell suspensions resulting from mechanical homogenization ($n = 33$) (black) or after collagenase treatment ($n = 33$) (light grey). The viral copy numbers between the two preparations were significantly different for B19V ($P < 0.001$), while not significant for EBV ($P > 0.05$). Bottom panels: median copy numbers of B19V DNA ($n = 33$) (**c**) and EBV DNA ($n = 24$) (**d**) in T-, B- and monocytic cells following homogenization (black) or collagenase digestion (light grey). The viral loads in B cells were statistically significant between preparations for B19V ($P < 0.001$) while not significant for EBV ($P > 0.05$). Error bars indicate 95% confidence intervals and were determined by normal approximation. Statistical significance was calculated with Mann–Whitney *U* asymptotic sig. (two-sided test).

IgD$^-$, CD27$^+$ IgD$^{low}$, CD27$^+$ IgD$^{high}$, and CD27$^-$ IgD$^+$ (Supplementary Fig. 5a). Both the viral loads and cell counts in the fractions were low; the only exception being two children (previously mentioned with the highest copies in the cohort) whose viral DNA copies were one log higher in the CD27$^+$ IgD$^{high}$ than in other sub-populations (Supplementary Fig. 5b,c).

**Mechanism of B19V entry into B cells.** ADE facilitates B19V entry into monocytes[15] and endothelial cells[16], although in the latter the process is mediated by proteins of the complement system.

ADE was tested as a possible entry route in two B-cell lines, Raji and GM12878, and on primary tonsillar B cells. These cells were confirmed to express by flow cytometry the Fc-receptor CD32 and the B19V receptor globoside (Supplementary Fig. 4a–c).

The cells were cultured in the presence of either B19V IgG-positive or -negative human sera, as well as of total purified IgG preparations. Virus internalization was followed up both by flow cytometry and confocal microscopy using fluorescent B19 virus-like particles (VLP, non-infectious), consisting of the major capsid protein VP2 (VP2-VLP-AF647). The cells were trypsinized before analysis to trim un-specific signals by non-internalized particles.

In GM12878 and Raji cells, the VLP uptake was detected only in the presence of B19V-IgG$^+$ sera at dilutions down to 1:50 or 1:100, respectively, or of corresponding total purified IgGs at $\geq 0.5$ mg ml$^{-1}$ (Fig. 2a–d). Raji cells showed higher and faster uptake efficiency than GM12878 cells.

A reduction in the fluorescence signal of 45.7% in Raji ($P < 0.05$, Student's *t*-test) and of 42.1% in GM12878 ($P = 0.05$, Student's *t*-test) cells was observed upon blocking of the CD32 molecule with specific antibodies, thus confirming internalization via Fc receptor (Fig. 2e,f). The VP2-VLP-AF647 uptake was also analysed in the presence of B19V-IgG$^+$ serum before or after heat-inactivation to evaluate the role of complement. However, no significant difference was observed (Fig. 2c,d).

ADE was also tested in primary tonsillar B cells from two B19V-seronegative individuals, with capsid uptake detected exclusively in the presence of specific B19V IgGs at concentrations down to 0.5 mg ml$^{-1}$ ($P < 0.01$, Student's *t*-test; Fig. 3a,b). Similar results were obtained with the monocytic cell line U937, in which B19V-ADE was first described, although via the CD64 receptor, thus confirming the suitability of our method to test this mechanism (Fig. 3c).

B19V-capsid internalization was substantiated in both B-cell lines by confocal microscopy (Fig. 4a,d). The VP2-VLP-AF647 capsids localized within early endosome antigen 1 (EEA1)-positive endosomes at the cell periphery (Fig. 4b,e). Line profile analysis confirmed that the VLP and EEA1 localizations were similar in both cell lines (Fig. 4c,f). However, the amount of VLPs in endosomes was higher in the Raji cells than in GM12878 cells. Altogether, our findings suggest that B19V-VLPs were able to enter the cells via an endocytic mechanism.

ADE was evaluated moreover using native *ex vivo* B19 viruses from a high-titre viremic plasma at 10 particles per cell in the presence of 1 mg ml$^{-1}$ B19V-positive or -negative total purified IgGs (Fig. 5a,b). In the presence of virus-specific antibodies a significant increase in viral DNA copy numbers was observed ($P \leq 0.01$, Student's *t*-test). Moreover, a reduction in virus internalization of 99.6% in Raji and of 85.6% in GM12878 cells was measured upon CD32 receptor blockage ($P < 0.01$, Student's *t*-test, Fig. 5c,d).

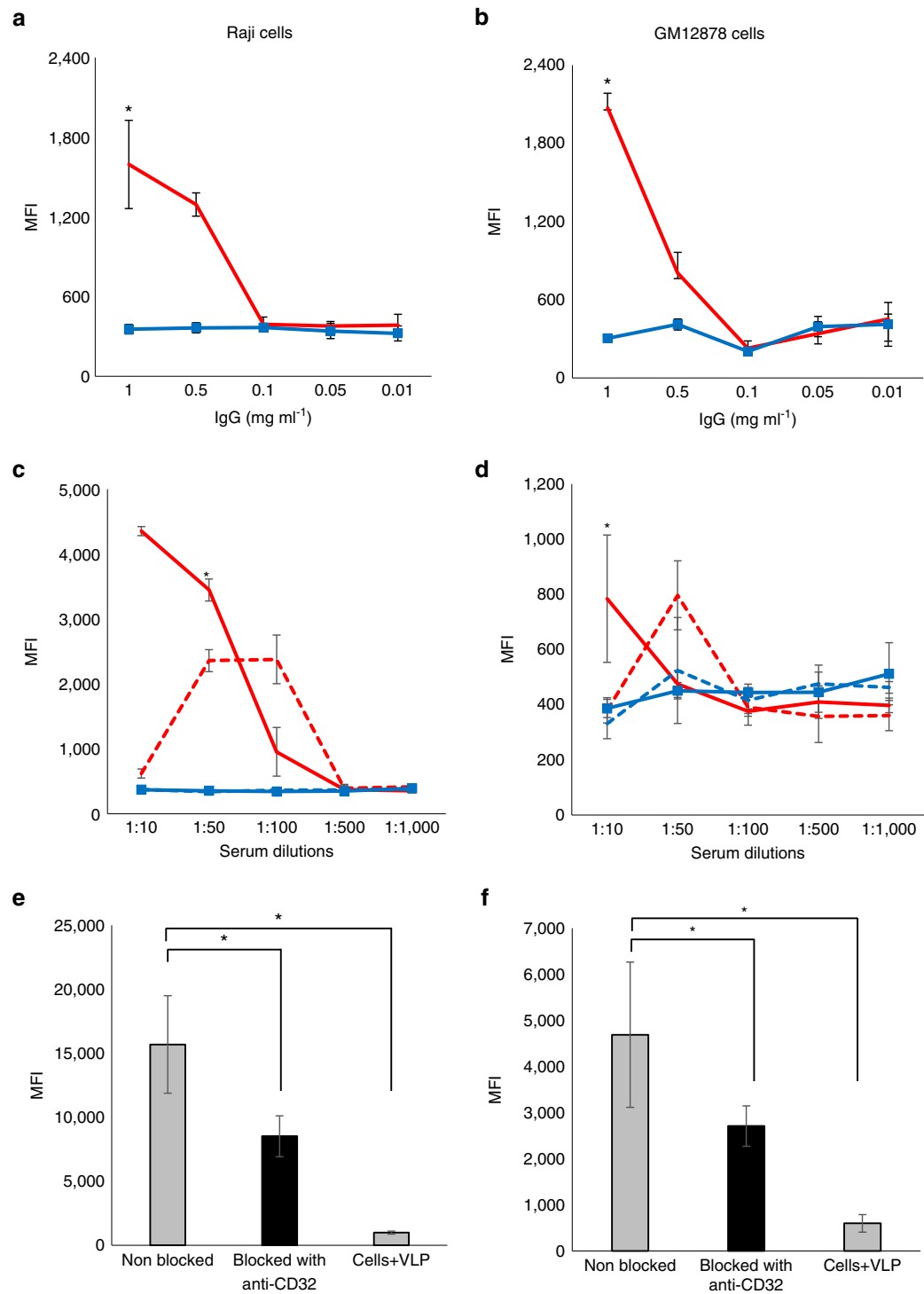

**Figure 2 | Antibody-dependent uptake of VP2-VLPs in B cell lines.** The mean fluorescence intensity (MFI) of VP2-VLP-AF647 was measured by flow cytometry in Raji (left column) and GM12878 cells (right column) after incubation with total purified B19V-positive (red line) or -negative IgGs (blue line, squares) (**a**,**b**). Pooled sera from B19V-IgG[+] individuals either heat inactivated (red solid line) or non-heat inactivated (red dotted line) or pooled sera from B19V-IgG[-] individuals either heat inactivated (blue solid line, squares) or non-heat inactivated (blue dotted line) were used (**c**,**d**). * Represents statistical significance ($P < 0.001$) between positive and respective negative controls at a given concentration. The effect on viral uptake was evaluated before (grey bars) and after (black bars) blocking with an anti-CD32 antibody (**e**,**f**). The difference was statistically significant (Raji $P < 0.001$, GM12878 $P = 0.05$). Error bars represent s.d. (of four replicates). Statistical significance was calculated using Student's *t*-test.

## Discussion

We searched for cell type(s) accounting for lifelong B19V DNA tissue persistence. Hypothesizing that long-lived lymphoid cells may be suitable hosts, we studied the distribution of B19V DNA in tonsillar B, T and monocyte cell lineages.

To release long-lived cells that are associated to connective tissue-rich areas of the tonsil[18], we employed a two-step preparation process of cell suspensions involving mechanical homogenization followed by collagenase digestion. We found that the B19V-DNA loads were significantly higher in the CD19[+]

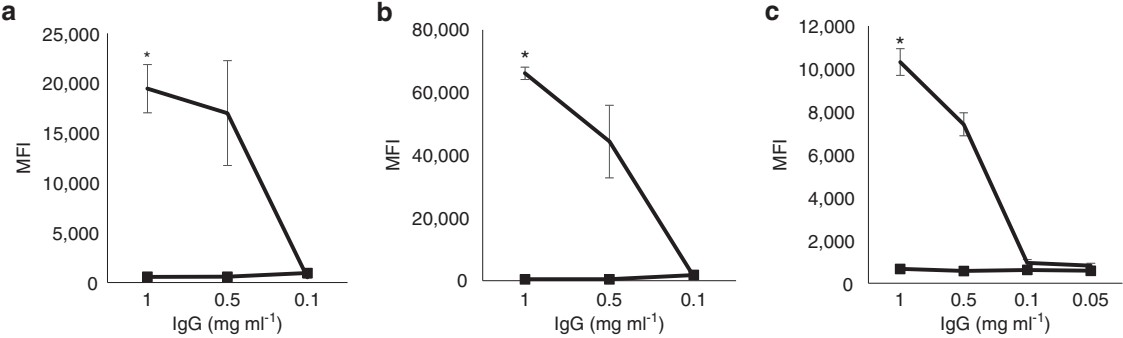

**Figure 3 | Antibody-dependent enhancement of VP2-VLPs in primary B cells and in a monocytic cell line.** The mean fluorescence intensity (MFI) of VP2-VLP-AF647 was measured by flow cytometry in tonsillar B cells from two seronegative individuals (**a**,**b**) as well as in U937 cells (**c**) after incubation with B19V-IgG$^+$ (black solid line) or B19V-IgG$^-$ (black solid line, squares) total purified preparations. Error bars represent standard deviation (of four replicates). *Represents statistical significance ($P < 0.001$) between positive and negative IgGs at a given concentration as calculated by Student's $t$-test.

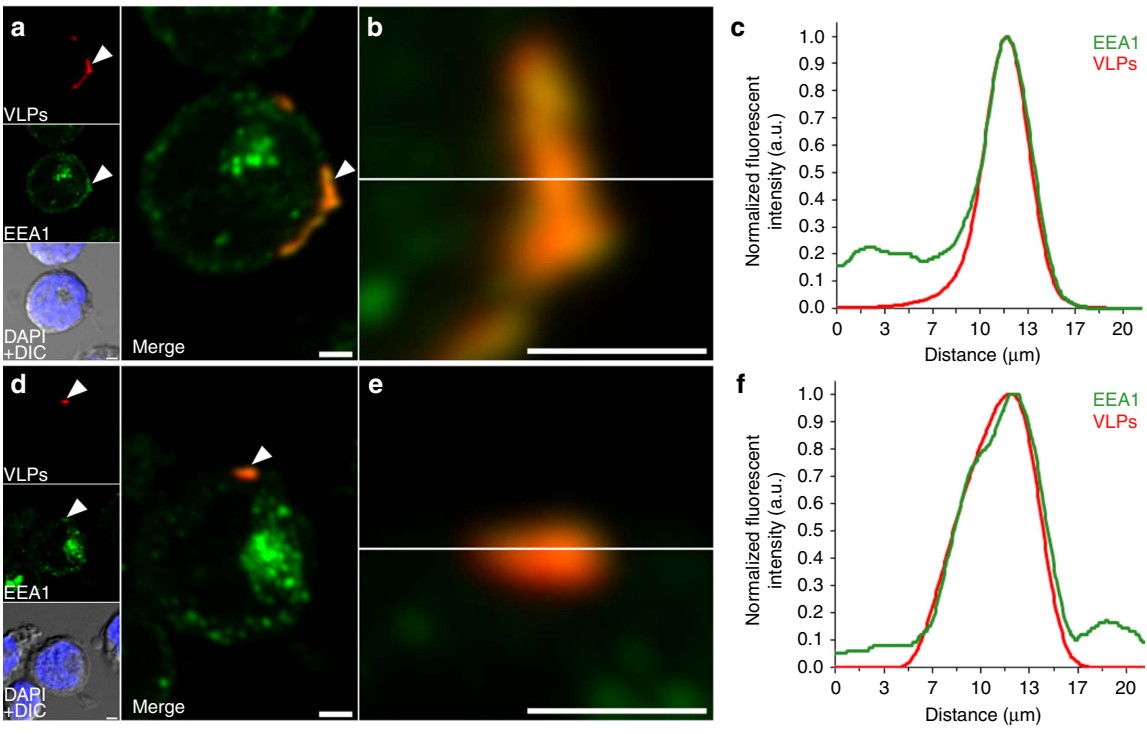

**Figure 4 | Association of VP2-VLPs with EEA1-positive endosomes.** Maximum intensity projections of serial confocal optical sections through Raji (**a**) and GM12878 cells (**d**) incubated overnight with VP2-VLP-AF647 (red) and treated with trypsin. White arrows represent enhanced cytoplasmic VLP localization in EEA1-positive endosomes (green). Differential interference contrast (DIC) merged with DAPI (blue) images are shown. Areas of localization (yellow) (**b**,**e**) and normalized fluorescent intensity profiles in zoomed areas (**c**,**f**) indicate VLP internalization. Scale bars, 2 μm.

cells released by collagenase, thus suggestive of persistence in long-lived B cells. In contrast, the viral loads of another B-cell infecting virus, Epstein-Barr[19], showed no comparable difference among preparations, thus not only precluding a methodical bias as a reason for the results but also reflecting the comparatively different life cycles of these two viruses. Indeed, EBV regularly reactivates and constantly infects *de novo* B cells as well as several other susceptible cell populations[20].

We then sorted the CD19$^+$ cells of individuals with the highest B19V-DNA copy numbers based on their expression of CD27 (memory) or IgD (naive) surface markers. However, the overall low B19V-DNA quantities hampered any clear resolution on the subtype of B cells harbouring the viral DNA. Even though not representative of the cohort, substantial data

were derived solely from two individuals, ages 6 and 8, with the highest B19V-DNA copy numbers, in whom the viral DNA was found predominantly in a cluster of B cells expressing high levels of both markers. It has been suggested that in young children, germinal center-derived memory cells are preferentially of IgM rather than of IgG type[21]. Thus, this B-cell subpopulation may represent IgM$^+$CD27$^+$IgD$^+$ cells which have been demonstrated to be of memory type and to derive from common germinal center reactions as clonally related, class switched, IgG$^+$ memory B cells[22]. Of note, the highest B19V-DNA copies in these two individuals were found exceptionally in the monocyte fraction; yet they were B19V-IgM negative and non-viremic, suggesting non-acute albeit relatively recent primary infection.

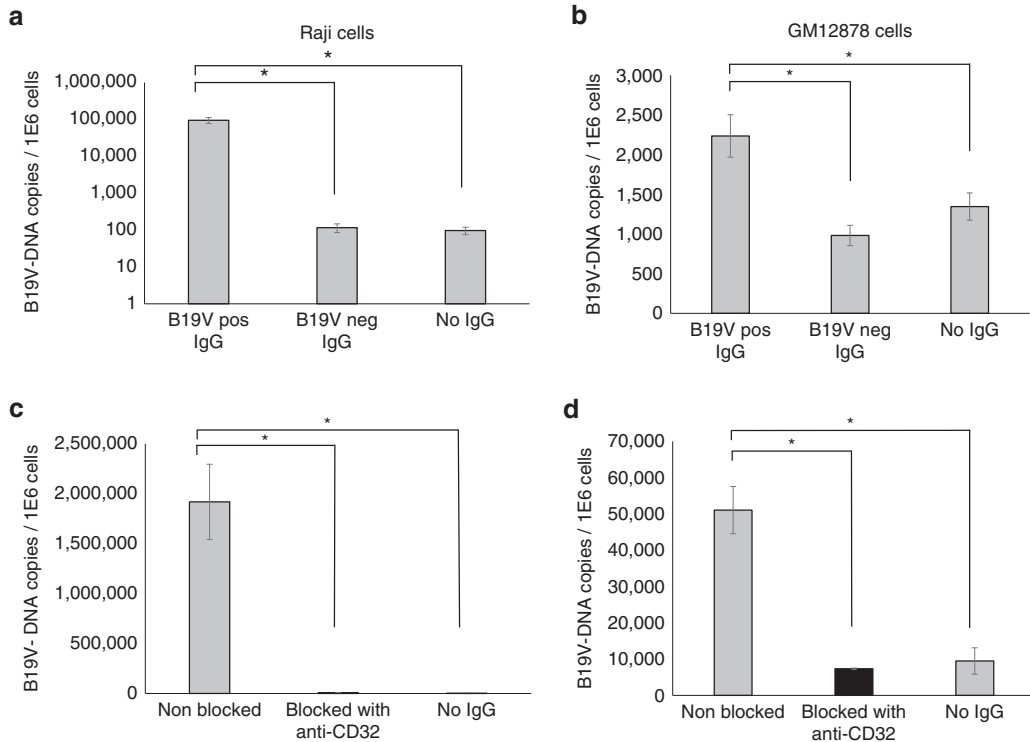

**Figure 5 | Antibody-dependent uptake of native B19V in B cell lines.** ADE was tested using viremic plasma in Raji (left column) and GM12878 (right column) cells in the presence of total purified B19V-positive or -negative IgGs (**a,b**). The effect on viral uptake was evaluated before (grey bars) and after (black bars) blocking with an anti-CD32 antibody (**c,d**). Error bars represent s.d. (of three replicates). *Represents statistical significance ($P < 0.001$) between positive and respective negative controls, as calculated by Student's $t$-test.

We isolated CD19[+] cells from venous blood of B19-seropositive individuals, but found the blood-derived cells to be B19V-DNA negative. This may reflect the rapid turnover of circulating B cells (from a few days to 5–6 weeks)[23] and/or their difference in phenotype as compared to tonsillar B cells[24,25].

Even though the definitive subset of the B19V-DNA-laden B cells remains to be determined, the specific genotype of B19V found provided compelling evidence of its persistence in long-lived cells. Indeed, out of the three described viral genotypes, the type 2 disappeared from circulation around 1970 (refs 3,4). In our cohort, four individuals older than 45 years carried this extinct virus type within their B cells, thus alluding to the longevity of these cells. Indeed, since the B19V-DNA copy numbers in tissue wane with time and there are no detectable traces of the virus in blood or secretions other than during recent infection, it is unlikely that continuous viral replication or re-infection could explain the finding in B cells of a pathogen that circulates no more[11].

Although there are some isolated case reports of B19V-type 2 viremia, particularly in immunocompromised individuals[26,27], the extreme rarity of such events rules against regular replication or re-activation of the persistent virus. Also, by molecular data B19V replication has been reported to be slow or absent during persistence[28] and no evidence exists of B19V integration into the host genome.

Thus, the common occurrence of the extinct type 2 B19V in our ageing individuals suggests for these very B cells a survival time of at least 40 to 60 years. This indicates that immunological memory is not only maintained in the absence of antigen re-exposure[29,30] but that it is due to long-term survival of original clones.

While the presence of B19V capsid proteins has been reported in lymphoid cells of inflamed synovial tissue sections[31,32], the mechanism of virus uptake into B cells has never been examined. We demonstrated the entry to be antibody mediated and CD32 dependent using both synthetic VLPs and native viruses, in line with a previous finding in monocytes[15].

The CD32 receptor plays an important role in the down-regulation of B-cell responses and its deficiency or reduced expression correlates to prolonged humoral or autoimmune reactivity[33,34]. Analogously, B19V-related arthropathies are allegedly due to IgG deposition[1] and this virus has been proposed to play a role in triggering or exacerbating auto-immune diseases[35]. Thus, our finding may provide a basis for the prolonged B-cell activation and the sustained polyclonal antibody production so characteristic of B19V acute infection[35].

Moreover, B19V seropositivity has been correlated to specific patterns of gene promoter DNA methylation in children's B-cell acute lymphoblastic leukaemia[36]. Even without evidence of causality, it might be relevant to expand on the mechanisms by which B19V could modulate the affected B cells and whether it might play a role in the development of proliferative disorders.

In the present study, we (i) demonstrated a cellular site for B19V lifelong tissue persistence, (ii) provided a marker for the longevity of B cells based on the detection of a no longer circulating virus type and (iii) documented antibody-dependent enhancement as the uptake mechanism.

## Methods

**Tonsil collection and preparation.** The right palatine tonsil and a serum sample were collected from 77 patients each, who underwent tonsillectomy/tonsillotomy for chronic tonsillitis or tonsillar hypertrophy during April-May 2015 at the Department of Otorhinolaryngology-Head and Neck Surgery of Helsinki University Hospital. The patients' ages ranged from 2 to 69 years (mean 22). The Ethics Committee of Surgery of Helsinki and Uusimaa Hospital District

approved the study protocol. The patients provided written consent before entry into the study.

The tonsils were collected immediately after surgery into tubes containing PBS and were kept on ice through the preparations. They were cut into smaller pieces with disposable scalpels and two separate tonsillar cell suspensions were prepared by a two-step process consisting of: (i) mechanical homogenization with a syringe plunge; (ii) collagenase digestion of the residual tonsillar tissue (from the previous step) with 0.1 mg ml$^{-1}$ Liberase TL (Roche) at $+37\,^{\circ}C$ for 30 min. Each cell suspension was washed with 50 ml PBS and filtrated through a 70 µm nylon mesh (Corning Life Sciences). Henceforth, each of the cell suspensions were treated and analysed independently.

The B, T and monocyte/macrophage cell fractions were enriched using anti-CD19, -CD3, -CD14 magnetic beads, respectively, following manufacturer's protocol (Invitrogen). The purity of the cell fractions was tested from six individuals by incubation with a 1:100 dilution of anti-human CD19-FITC (#MHCD1910, Invitrogen), CD3-PE/cy5.5 (#ab124052, Abcam) and CD14-PE (#345785, BD Biosciences) on ice for 30 min and analysed on a BD Accuri C6 Flow cytometer.

Anti-mouse Ig, k/Negative control compensation particles set (#552843, BD Biosciences) as well as corresponding fluorescent-labelled isotypes served as controls.

**Lymphoid cell isolation from venous blood.** Venous blood samples were collected from one B19V-seronegative and seven seropositive staff members. Informed consent was obtained form all study subjects and the study protocol was approved by the Ethics Committee of the Helsinki and Uusimaa Hospital District. The blood was collected using BD vacutainer CPT tubes (BD Biosciences) and the PBMCs separated as instructed by the manufacturer. From this, the CD19$^+$ cells underwent magnetic cell separation as specified in the previous section.

**DNA extraction and quantification.** The DNA from tonsillar cells was extracted with the KingFisher Duo purification system (12-sample format) using the KingFisher Cell and Tissue DNA Kit (both Thermo Scientific) according to manufacturer's instructions. PBS ($n = 2$) was used as a negative control during the extractions.

B19V DNA was quantified with (i) Pan-B19V qPCR amplifying a 154 bp region of the NS1 gene and confirmed with (ii) VP-qPCR targeting a 121-bp region of the VP1 gene, as previously described[11,37]. The Pan-B19V qPCR products of the 33 positive individuals were purified with Diffinity RapidTip2 (Sigma-Aldrich) and Sanger sequenced with corresponding qPCR forward and reverse primers at the sequencing unit of the Institute for Molecular Medicine Finland. The sequences were analysed with Bioedit v.7.2.5 (Ibis Biosciences) and compared to B19V reference sequences (GenBank accession numbers FN598217.1 and AY044266.2 for genotype 1 and 2, respectively) using BLAST.

EBV DNA was quantified with a qPCR modified from Aalto et al.[38], targeting the BALF5 DNA-polymerase gene. The EBV-qPCR reaction mix was composed of 12.5 µl of 2× Maxima Probe Master Mix (Thermo Scientific) with 0.03 µM of ROX passive reference dye, 0.25 µM of both forward and reverse primers, 0.17 µM of probe, 5 µl of template DNA and nuclease-free water in a final volume of 25 µl. The PCR program consisted of initial denaturation at 95 °C for 10 min, following 45 cycles of 95 °C for 15 seconds and 60 °C for 1 min. The analytical sensitivity was 10 copies/reaction.

To normalize the viral DNA copies to the cell number, the human single-copy gene RNase P was quantified as described[37].

For quantification and as a positive control, tenfold serial dilutions (1E6–1E1 copies per µl) of the following plasmids were used: (a) a near-full-length B19V genotype 1 genome without hairpin structures (cloned in-house)[39]; (b) RNase P -qPCR amplicon obtained from Dr. Janet Butel[40]; and (c) EBV-qPCR amplicon synthesized and cloned into the pIDTBlue vector by Integrated DNA Technologies (5′-CCCTGTTTATCCGATGGAATGACGGCGCATTTCTCGT GCGTGTACACCGTCTCGAGTATGTCGTAGACATGGAAGTCCAGAGGG CTTCCG-3′).

Each sample was analysed in duplicate.

All primers and probes used in this study were ordered from Sigma Aldrich. Detailed sequence information to be found in Supplementary Table 3.

**Serological studies.** The serostatus of the study subjects was determined with in-house B19V-IgG and -IgM EIAs as described[41,42]. In brief, for the B19V-IgG EIA, 45 ng per well of biotinylated B19-VP2-VLPs were coated onto streptavidin coated plates using sample Diluent (#6111040, Anilabsystems, Finland). The serum samples were diluted 1:200, tested in duplicate and detected with anti-human IgG horseradish-peroxidase conjugated (Dako).

B19V-IgM antibodies were measured using an in-house IgM capture EIA as described by Maple et al.[42] Microtiter plate wells (Costar, Corning) were coated with goat anti-human IgM (#55097, Cappel/MP Biomedicals) and blocked with 3% bovine serum albumin. The serum samples were diluted 1:200 in PBS-Tween (PBST), tested in duplicate, followed by addition of 10 ng per well of biotinylated B19-VP2-VLP. For detection, a 1:12,000 dilution of streptavidin horseradish-peroxidase conjugate (Dako) was added.

Both EIAs were developed by addition of o-phenylenediamine dihydrochloride (Dako) plus $H_2O_2$ and the absorbances read at 492 nm using a Labsystems Multiscan EX (Thermo Fisher).

**Cell sorting of primary tonsillar B cells.** Collagenase-treated tonsillar cell suspensions were stored at $-196\,^{\circ}C$ after homogenization. Samples from 12 individuals were washed twice with PBS after thawing to remove excess DMSO, resuspended in 10% FBS-PBS solution for 10 min at 4 °C and stained with 1:100 dilution of anti -CD19-PE (#ab1168, Abcam), -CD27-FITC (#ab30366, Abcam), -IgD-APC (#561303, BD Pharmingen) on ice in the dark for 30 min, washed three times with PBS and immediately sorted in a BD Influx flow cytometer cell sorter.

As controls and for compensation, unstained cells, single antibodies and corresponding fluorescent-isotype controls plus anti-mouse Ig, k/Negative control compensation particles (BD Biosciences) were used.

The sorting was based on gated CD19$^+$ cells and four cell subpopulations were collected in PBS according to the expression of either CD27 or IgD markers as: CD27$^+$ IgD$^-$, CD27$^+$ IgD$^{low}$, CD27$^+$ IgD$^{high}$ and CD27$^-$ IgD$^+$. The B19V-DNA copies were quantified from each of these subpopulations as specified above.

**Cell lines.** Two human B-cell lines (GM12878 and Raji (GM04671) cells, both from Coriell Institute) and a monocytic cell line, U937 (ATCC CRL-1593.2), were cultured in RPMI 1640 + GlutaMAX-I medium (Gibco), 1% penicillin-streptomycin (Sigma-Aldrich) and complement-inactivated 10% FBS (Gibco) at $+37\,^{\circ}C$ and 5% $CO_2$.

Primary tonsillar B cells from two seronegative individuals of the study cohort were isolated with anti-CD19 magnetic beads from frozen total cell suspensions of mechanically homogenized preparations and released with DETACHaBEAD as instructed by manufacturer (Invitrogen).

**Serum pools and purified human immunoglobulins.** Two negative pools and a positive pool of sera ($n = 5$, each) were tested before and after complement inactivation.

Total immunoglobulin G was purified from a B19V-IgG negative pool ($n = 4$), using the HiTrap Protein A HP column (GE Healthcare) according to manufacturer's instructions. The final IgG concentration was measured with the Pierce BCA protein assay kit (Thermo Fisher Scientific) following manufacturer's protocol.

A total purified IVIG preparation (Gammagard S/D, Baxter), of known B19V reactivity as per in-house EIA, served as positive control.

Both purified IgG preparations were complement inactivated.

**Antibody-dependent enhancement.** The effect of B19V specific IgGs in the internalization of VP2 VLP was tested on GM12878, Raji and U937 cell lines as well as on primary tonsillar B cells from two B19V-seronegative individuals.

B19-VLPs consisting of the VP2 capsid protein were produced with the BaculoGold Baculovirus Expression System (BD Biosciences) and conjugated to Alexa Fluor 647 using a protein labelling kit (Thermo Fisher Scientific) according to manufacturer's protocol. The immunoreactivity and purity of the VP2-VLP-AF647 were assessed by in-house EIA and SDS-PAGE followed by fluorescence imaging in Odyssey (LI-COR), respectively.

In each well, 1.50E5 of cells were cultured with 100 ng of VP2-VLP-AF647 and incubated with either purified human IgGs or pooled sera. The purified human IgGs were used at final concentrations of 1, 0.5, 0.1, 0.05 or 0.01 mg ml$^{-1}$ while the serum pools, with or without heat inactivation, were tested at dilutions of 1:10, 1:50, 1:100, 1:500 or 1:1,000. The uptake was evaluated at different time points up to 16 h, which was then used as the standard time point for this ADE experiments. All conditions were tested in four replicates in the same plate and in at least two separate occasions. Cells only and cells combined with VP2-VLP-AF647 (without antibodies) were used as controls through the studies.

ADE was also tested in the B-cell lines using live B19V from acute infection plasma. To this end, 5E6 cells were incubated with 10 B19V particles per cell for 40 h together with 1 mg ml$^{-1}$ of either B19V-positive or -negative total purified human IgGs. All tests were carried out in triplicate and on two or more separate occasions.

Following overnight incubation, the cells were washed with cold PBS, incubated with 0.25% trypsin-EDTA (Gibco) for 15 min at $+37\,^{\circ}C$, washed three times with cold 20% FBS-PBS and finally resuspended into PBS.

The efficiency of trypsin digestion was assessed by flow cytometry via expression of CD19-FITC (#MHCD1901, Invitrogen) and CD71-FITC (#555536, BD Pharmingen) as well as of cell viability with propidium iodide (Invitrogen).

After trypsin digestion, ADE was determined in the VP2-VLP-AF647 treated cultures by measuring the mean AF647 fluorescence signal with BD Accuri C6 flow cytometer (BD Biosciences).

With live B19V, DNA from the cell cultures was extracted with the DNA Blood Mini Kit (Qiagen) according to the manufacturer's instructions for viral DNA, and the B19V and RNase P copies were quantified by the respective qPCRs (Pan-B19V and RNase P), as specified in DNA extraction and quantification.

**Blocking of Fc-receptor.** A total of 3E6 Raji or GM12878 cells were pre-incubated with $20\,\mu g\,ml^{-1}$ of anti-CD32 (#551900, BD Biosciences) for 1 h at $+37\,°C$, then distributed to 1E5 cells per well and incubated with 100 ng of VP2-VLP-AF647 overnight. Cells alone and cells with VLPs only were used as controls.

In the assays with live B19V, 2E6 Raji or GM12878 cells were pre-incubated with $40\,\mu g\,ml^{-1}$ of anti-CD32 (BD Biosciences) for 1 h at $+37\,°C$, and then incubated with the live virus at 10 B19V particles per cell and 5E5 cells per well, in the presence of $1\,mg\,ml^{-1}$ of purified IgGs. After 40 h, the cells were washed and trypsinized as described above.

**Confocal microscopy.** VLP internalization and co-localization with intracellular structures was assessed by confocal microscopy.

After measurement of AF647 by flow cytometry, the cells were spread and air-dried on Zeiss high-performance cover slips and fixed with 4% paraformaldehyde (PFA; 20 min at room temperature). Early endosomal vesicles were detected with a 1:150 dilution of EEA1 mouse monoclonal antibody (#610457, BD Biosciences), and the DNA stained during embedding with DAPI (4'-6-diamidino-2-phenylindole; Thermo Fisher Scientific), containing ProLong antifade compound. Imaging was done using an Olympus FV-1000 inverted confocal microscope (Olympus) with the UPLSAPO $\times 60$ oil-immersion objective (numerical aperture = 1.35). Alexa 647 was excited with a 633 nm He-Ne laser and the fluorescence collected with a BA650IF long-pass filter. Alexa 488-conjugated anti-rabbit IgG was excited with a 488 nm argon laser and its emission detected with a 500 to 555 nm band-pass filter. DAPI was excited with a 405 nm diode laser and detected with a 425 to 475 nm band-pass filter. For 3D image stacks, $320 \times 320$ pixels were collected from an appropriate depth depending on the sample. Pixel resolution was adjusted to $\sim 70$ nm per pixel in $x$ and $y$ dimensions, and to 150 nm in the $z$ dimension. Iterative deconvolution was performed with a signal-to-noise ratio set at 5 and a quality threshold set at 0.05, using Huygens Essential software (SVI). Image analysis was done with ImageJ[43].

**Immunolabelling of cell surface markers.** The expression of Fc and globoside receptors was assessed on cultured Raji and GM12878 cells, as well as on CD19$^+$ cells isolated from tonsillar tissue and venous blood. Briefly, 1E6 cells were incubated with $5\,\mu g\,ml^{-1}$ (1:100 dilution) of either mouse anti-human-CD32 (#551900, BD Pharmingen) or -CD64 antibodies (#55525, BD Pharmingen) on ice for 30 min, washed three times with PBS and incubated with $5\,\mu g\,ml^{-1}$ (1:100 dilution) of goat anti-mouse IgM/IgG-FITC secondary antibody (#555988, BD Biosciences) for 30 mins on ice and washed three times before analysis on BD C6 Accuri.

The staining of globoside was done in a similar fashion using a 1:50 dilution of rabbit polyclonal anti-globoside GL4 (#1960, Matreya LLC) and $5\,\mu g\,ml^{-1}$ (1:100 dilution) of goat anti-rabbit-FITC conjugated secondary antibody (#554020, BD Biosciences).

The antibody concentrations were optimized before analysis and both unstained cells as well as cells stained with secondary antibody only were used as background controls.

**Statistical analysis.** The median CI was calculated with normal approximation using OpenEpi v3 open source calculator. SPSS v22.0 was used to calculate the Mann-Whitney $U$ asymptotic sig. (two-sided test) and Pearson's $\chi^2$-test. The difference in mean fluorescence intensity in the ADE experiments was calculated with Student's $t$-test in RStudio (v3.2.4).

**Data availability.** All the relevant data are available from the corresponding author upon request.

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

## Acknowledgements

We thank Petriina Mannelli for help with tissue collection, Visa Nurmi and Jussi Hepojoki for providing serum samples and fluorophore-labelled VLPs, Lassi Palmujoki for help with confocal imaging, Prof. Jukka Pelkonen for scientific guidance, Pieta Mattila for comments on the manuscript, Kalle Kantola for technical support with figures and the Biomedicum FACS core facility where the flow cytometry and cell sorting were performed. This work was funded by the Academy of Finland (grant 1257964), the Jane and Aatos Erkko Foundation (K.H. and M.V.-R.), the Sigrid Jusélius Foundation, the Helsinki University Hospital Research & Education Fund, the Finnish-Norwegian Medical Foundation, the Medical Society of Finland (FLS), the Research Funds of the University of Helsinki and the Life and Health Medical Grant Association.

## Author contributions

L.P., M.T., E.M., L.H., M.F.P. performed the experiments; L.-M.A. and T.I. provided serum and tissue samples and clinical data; L.P., M.T., E.M., M.F.P. performed the data analysis; K.H., M.S.-V., M.V.-R., M.F.P. contributed to the study design; all authors participated in interpretation, manuscript writing and editing.

## Additional information

**Competing interests:** The authors declare no competing financial interests.

