## [Peer Review File · Nature Communications]

Reviewers' comments:

Reviewer #1 (Remarks to the Author):

Review of Manuscript "Longevity of B cells revealed via an extinct type of human parvovirus B19" by Lari Pyöriä and coworkers.

Whereas the highly cell type specific productive B19V infection in certain stages of erythroid precursor cells and lifelong persistence after primary infection has been demonstrated for many years, the target cells for establishing this lifelong persistence have not been characterized yet.

In their present study the authors show several lines of evidence that are highly suggestive of long-lived memory B cells as sites responsible for B19V persistence. Most importantly, by sequencing of PCR products of a selected region of the B19V NS1 gene they identified in B cells a B19V genotype (genotype 2), which has been shown to have disappeared from circulation about 45 years ago. Furthermore, they were able to detect much higher B19V DNA loads in B-cells when they made use of a two-step preparation process shown to enrich for cells associated with connective tissue-rich areas, which also includes memory cells. In addition to these in vivo studies on the cell types harboring persistent B19V genomes the authors then present in vitro data on the possible entry mechanism of B19V into B cells, which they show to be B19V-antibody dependent.

Whereas the principal findings presented in the study are of major interest for virologists and the experimental data also look very sound and are of high quality, several substantial points should be addressed in a revised version of the manuscript:

1) The detection of persistent B19V genomes is based solely on a small region within the NS1 gene. To assure, that the B cells of interest really contain large parts of the B19V genome and not just small B19V subfragments, the NS1-specific PCR should be complemented by a second PCR detecting a distant region of the B19V genome, e.g. in the VP gene.

2) The data set for the distribution of B19V genotypes seems to be rather small (only the PCR products from 6 samples were actually sequenced) and should be extended.

3) In view of the known entry mechanisms of B19V and the importance of the VP1 unique (VP1u) region in this process, the use of virus-like particles (VLP) containing only VP2 to study the uptake mechanism in B cells seem to be rather artificial. An antibody-dependent mechanism (ADE) as suggested by the authors can also be studied using B19V from patient material and such experiments would strongly strengthen the present data.

4) (related to point 3) In view of the rather artificial system, the experiments dealing with the antibody-dependent entry mechanism into B cells seem to be overrepresented in the manuscript.

5) The authors failed to demonstrate the exact subset of B cells (naive, mature cells) responsible for establishing persistent B19V infection and should discuss on that. The two young patients harboring high viral loads in the monocyte fraction, which are discussed in detail, seem to be rather non-representative for the complete set of patients.

Reviewer #2 (Remarks to the Author):

The manuscript by Pyöriä et al. describes a comprehensive set of experiments towards the identification of B cells as possible reservoirs for human parvovirus B19. The strengths of the manuscript include the following: (i) The authors are established parvovirus B19 investigators who have published on B19 persistence in multiple tissue types; (ii) The sample size of the study population is very good at 77; and (iii) The determination of B19 sero-positivity of matched sera enhances the validity of the study, as does comparisons with EBV. However, there are significant weaknesses as well. For example: (i) Some of the data shown are not particularly novel; and (ii) Some of the experimental approaches used are not ideal. The idea of memory B cells as long-term reservoirs is intriguing and an elegant idea; however, B cells have previously been shown to harbor B19 DNA, RNA, and protein in lymphoid tissues (Ref. 28). The authors were also not able to definitively determine the phenotype of the B cells as memory in nature. The antibody-dependent endocytosis (ADE) mechanistic work is only somewhat novel in that ADE for B19 with both bone marrow cells and monocytes was documented more than a decade ago (Ref. 7), and with endothelial cells a couple of years ago (Ref. 8), although the current studies are focused specifically on B cells. These data actually detract from the overall theme of the paper. The truly interesting aspect of persistence of the virus in B cells, and the sequence analysis and comparison should take center stage. In fact, there is really no mention of the methodology or any data shown regarding the sequencing and comparisons to known sequences. This gets lost in the ADE work.

One also questions the overall validity of using the virus-like particle (VLPs) composed VP2 only, rather than the actual virus, as was done in previous studies with monocytes and endothelial cells.

It appears that all PATIENT samples were obtained from a single geographic location, and if so, how likely is it that their conclusions can be extrapolated to include the global population? One also does not get an idea of what the results might be with NORMAL human tissues [I realize that it might not be possible to obtain tonsillar tissues from normal volunteers]. At the very least, these points should be discussed. The manuscript also needs major revisions to methodology and data regarding sequencing and analysis/comparisons.

Reviewer #3 (Remarks to the Author):

The human pathogen parvovirus B19 (B19V) preferentially replicates in erythroid progenitor cells leading to short lived viraemia, yet B19V DNA can also be found in a wide range of tissues for many years following primary infection, even in asymptomatic immunocompetent individuals. While these findings suggest that B19V can establish lifelong persistence, to date little is known about the potential sites of virus persistence or mechanisms of virus entry into target cells.

In the current study, the authors posit that B19V may persist in long lived memory B cells,

as is the case for Epstein-Barr virus, a widespread B lymphotropic herpes virus. To test this hypothesis, the authors examined the distribution of B19V DNA in B cells, T cells and monocytes fractionated from tonsillar cell suspensions using a highly sensitive Q-PCR assay. The results show that B19V DNA is frequently detectable in tonsil tissues, with the virus loads generally being highest in the B cell population. However attempts to further characterise the precise B cell subset carrying B19V DNA were unsuccessful. The authors also explored possible mechanisms of virus entry by exploiting an in vitro infection system using fluorescently labelled virus-like particles carrying the major capsid protein VP2. The data show that B cell lines and primary B cells can internalise these VLPs via the Fc gamma receptor CD32 in an antibody dependent mechanism.

In summary, this study describes significant new findings about B19V persistence and entry that will be of interest to the parvovirus community. However, while the VLP experiments appear robust, the data about the distribution of B19V DNA in different tonsillar subsets is less convincing and the hypothesis that B19V persists in B cells is not proven. Furthermore, the study fails to address several important questions: is B19V DNA detectable in circulating B cells, how is B19V DNA maintained long term in B cells and how does the virus gain access to other cell types in the apparent absence of virus replication?

Major points

1. The differences in B19V genome loads between mechanically homogenised and collagenase digested tonsil samples are intriguing but not fully explained (Figure 1a, 1c). For example, have the authors looked for differences in the proportion of B cells, T cells and monocytes that are recovered using these two methods? Are other cell types such as plasma cells or epithelial cells present in one method but not the other? It is important that these issues are experimentally addressed and discussed to explain why the B19V loads are significantly higher in the collagenase digested samples. It is interesting that the two methods give similar results for EBV (used here as a comparator). Given that EBV selectively infects B lymphocytes, these data would imply that B19V DNA cannot be exclusively present in B cells. Finally, the authors should be aware that certain tonsillar B cell subsets, such as germinal centre centrocytes and centroblasts, are highly prone to apoptosis during isolation and this may also impact upon the results.
2. The present data do not definitively demonstrate that B19V DNA is present in B cells, as the extremely low virus loads (Figure 1c) could be entirely explained by the presence of B19V DNA in the 3-6% contamination with other cell types (line 86). Indeed the observation that EBV DNA was detectable not only in B cells, but also in the isolated T cells and monocytes (Figure 1d), strongly argues that cell contamination is a concern, thereby confusing the interpretation of the data. To convincingly demonstrate the presence of B19V DNA in B cells, the authors must use a direct approach, such as cell surface staining for CD19 combined with FISH for B19V DNA followed by FACS analysis.
3. Have the authors looked for B19V DNA in circulating B cells? If present, buffy coats (pre-screened for B19V DNA positivity) could be sorted to provide high numbers of different B cell subsets and then analysed for B19V DNA. This would help address the question of which

B cell types carry B19V DNA, which was hampered in the current work by the low cell yields.

4. If B cells are the reservoir for B19V, the authors should look for B19V DNA in patients receiving anti-CD20/Rituximab therapy.

5. In the experiments described in Figure 2, can the authors comment on the proportion of cells which became VLP positive.

Dr Andrew Bell
Institute of Cancer and Genomic Sciences
University of Birmingham, UK

Reviewers' comments:

Reviewer #1 (Remarks to the Author):

Review of Manuscript "Longevity of B cells revealed via an extinct type of human parvovirus B19" by Lari Pyöriä and coworkers.

Whereas the highly cell type specific productive B19V infection in certain stages of erythroid precursor cells and lifelong persistence after primary infection has been demonstrated for many years, the target cells for establishing this lifelong persistence have not been characterized yet. In their present study the authors show several lines of evidence that are highly suggestive of long-lived memory B cells as sites responsible for B19V persistence. Most importantly, by sequencing of PCR products of a selected region of the B19V NS1 gene they identified in B cells a B19V genotype (genotype 2), which has been shown to have disappeared from circulation about 45 years ago. Furthermore, they were able to detect much higher B19V DNA loads in B-cells when they made use of a two-step preparation process shown to enrich for cells associated with connective tissue-rich areas, which also includes memory cells. In addition to these in vivo studies on the cell types harboring persistent B19V genomes the authors then present in vitro data on the possible entry mechanism of B19V into B cells, which they show to be B19V-antibody dependent. Whereas the principal findings presented in the study are of major interest for virologists and the experimental data also look very sound and are of high quality, several substantial points should be addressed in a revised version of the manuscript:

1) The detection of persistent B19V genomes is based solely on a small region within the NS1 gene. To assure, that the B cells of interest really contain large parts of the B19V genome and not just small B19V subfragments, the NS1-specific PCR should be complemented by a second PCR detecting a distant region of the B19V genome, e.g. in the VP gene.

In compliance with the Referee's suggestion, the B19V-DNA positivity of the B cells from collagenase treated tonsillar tissues has now been confirmed with a second quantitative PCR amplifying a 121-bp region of the VP gene. Details on the method as well as on the results have been added to the manuscript accordingly (pages 4-5 and 13). A figure on the correlation between the copy numbers detected by the two qPCRs has been added to Supplementary Materials (Supplementary Figure S 1).

2) The data set for the distribution of B19V genotypes seems to be rather small (only the PCR products from 6 samples were actually sequenced) and should be extended.

Appreciating the Reviewer's suggestion, the products from the 33 B19V-qPCR positive samples have now been sequenced as noted in the revised manuscript (page 5 and 13). A table with each patient's age, year of birth, genotype and nucleotide identity to reference sequence has now been added to Supplementary Materials (Supplementary Table 1).

3) In view of the known entry mechanisms of B19V and the importance of the VP1 unique (VP1u) region in this process, the use of virus-like particles (VLP) containing only VP2 to study the uptake mechanism in B cells seem to be rather artificial. An antibody-dependent mechanism (ADE) as

suggested by the authors can also be studied using B19V from patient material and such experiments would strongly strengthen the present data.

The VP2-VLPs allowed us to demonstrate ADE by excluding some other mechanisms that could affect the virus entry and which cannot be ruled out by use of a live virus. Moreover, by fluorescence we were able to document and follow the VLP uptake in parallel using both flow cytometry and confocal microscopy.

However, the Reviewer's point is valid. Hence, ADE has now been tested in the two B cell lines (GM12878 and Raji) using viremic serum. The experimental details are given on page 17/18 and the results (page 8), showing a correlation between B19V load and B19V IgG- positive sera, are presented in a new illustration (Figure 5).

4) (related to point 3) In view of the rather artificial system, the experiments dealing with the antibody-dependent entry mechanism into B cells seem to be overrepresented in the manuscript.

We agree; and while strengthening the evidence (point 3), we have according to the Reviewer's wish kept to a minimum the discussion on ADE (page 11).

5) The authors failed to demonstrate the exact subset of B cells (naive, mature cells) responsible for establishing persistent B19V infection and should discuss on that. The two young patients harboring high viral loads in the monocyte fraction, which are discussed in detail, seem to be rather non-representative for the complete set of patients.

The exact subset of B cells harboring B19V proved indeed difficult to resolve due to the very low viral loads. As instructed, we now discuss this on page 9.

We agree with the Reviewer; the two young patients are certainly not representative of the whole cohort. This we now state more clearly in the Discussion (page 9)

Reviewer #2 (Remarks to the Author):

The manuscript by Pyöriä et al. describes a comprehensive set of experiments towards the identification of B cells as possible reservoirs for human parvovirus B19. The strengths of the manuscript include the following: (i) The authors are established parvovirus B19 investigators who have published on B19 persistence in multiple tissue types; (ii) The sample size of the study population is very good at 77; and (iii) The determination of B19 sero-positivity of matched sera enhances the validity of the study, as does comparisons with EBV. However, there are significant weaknesses as well. For example: (i) Some of the data shown are not particularly novel; and (ii) Some of the experimental approaches used are not ideal. The idea of memory B cells as long-term reservoirs is intriguing and an elegant idea; however, B cells have previously been shown to harbor B19 DNA, RNA, and protein in lymphoid tissues (Ref. 28). The authors were also not able to definitively determine the phenotype of the B cells as memory in nature. The antibody-dependent endocytosis (ADE) mechanistic work is

only somewhat novel in that ADE for B19 with both bone marrow cells and monocytes was documented more than a decade ago (Ref. 7), and with endothelial cells a couple of years ago (Ref. 8), although the current studies are focused specifically on B cells. These data actually detract from the overall theme of the paper. The truly interesting aspect of persistence of the virus in B cells, and the sequence analysis and comparison should take center stage. In fact, there is really no mention of the methodology or any data shown regarding the sequencing and comparisons to known sequences. This gets lost in the ADE work.

We are grateful to the Reviewer for pointing out the importance of performing the sequence analysis and comparison. Please see also our answer to question #2 of Reviewer 1. Sanger sequencing of the 33 B19 positive PCR products has been performed and annotated in the text (pages 5 and 13), and the data are included in the Supplementary Materials (Supplementary Table 1)

One also questions the overall validity of using the virus-like particle (VLPs) composed VP2 only, rather than the actual virus, as was done in previous studies with monocytes and endothelial cells.

We agree with the Reviewer. Please see our answer to question #3 of Reviewer 1, and the corresponding new data and experimental set up using natural ex-vivo viruses (pages 8, 17/18, Figure 3).

It appears that all PATIENT samples were obtained from a single geographic location, and if so, how likely is it that their conclusions can be extrapolated to include the global population? One also does not get an idea of what the results might be with NORMAL human tissues [I realize that it might not be possible to obtain tonsillar tissues from normal volunteers]. At the very least, these points should be discussed. The manuscript also needs major revisions to methodology and data regarding sequencing and analysis/comparisons.

The numerous scientific publications from Finland on B19V and its tissue persistence have been shown to be highly representative of the entire northern hemisphere.

With regard to normal human tissues, subjects with tonsillar hypertrophy as sole manifestation were amply included in our cohort. This condition, not being inflammatory, as a sample source may be the closest to healthy controls as ethically feasible.

Reviewer #3 (Remarks to the Author):

The human pathogen parvovirus B19 (B19V) preferentially replicates in erythroid progenitor cells leading to short lived viraemia, yet B19V DNA can also be found in a wide range of tissues for many years following primary infection, even in asymptomatic immunocompetent individuals. While these findings suggest that B19V can establish lifelong persistence, to date little is known about the potential sites of virus persistence or mechanisms of virus entry into target cells.

In the current study, the authors posit that B19V may persist in long lived memory B cells, as is the case for Epstein-Barr virus, a widespread B lymphotropic herpes virus. To test this hypothesis, the authors examined the distribution of B19V DNA in B cells, T cells and monocytes fractionated from

tonsillar cell suspensions using a highly sensitive Q-PCR assay. The results show that B19V DNA is frequently detectable in tonsil tissues, with the virus loads generally being highest in the B cell population. However attempts to further characterise the precise B cell subset carrying B19V DNA were unsuccessful. The authors also explored possible mechanisms of virus entry by exploiting an in vitro infection system using fluorescently labelled virus-like particles carrying the major capsid protein VP2. The data show that B cell lines and primary B cells can internalise these VLPs via the Fc gamma receptor CD32 in an antibody dependent mechanism.

In summary, this study describes significant new findings about B19V persistence and entry that will be of interest to the parvovirus community. However, while the VLP experiments appear robust, the data about the distribution of B19V DNA in different tonsillar subsets is less convincing and the hypothesis that B19V persists in B cells is not proven. Furthermore, the study fails to address several important questions: is B19V DNA detectable in circulating B cells, how is B19V DNA maintained long term in B cells and how does the virus gain access to other cell types in the apparent absence of virus replication?

Major points

1. The differences in B19V genome loads between mechanically homogenised and collagenase digested tonsil samples are intriguing but not fully explained (Figure 1a, 1c). For example, have the authors looked for differences in the proportion of B cells, T cells and monocytes that are recovered using these two methods?

We appreciate the Reviewer's comment. The proportion of T and B cells recovered by the two methods is similar. Monocytes, on the other hand, are proportionately enriched in the cell preparations obtained in the absence of collagenase.

Are other cell types such as plasma cells or epithelial cells present in one method but not the other? It is important that these issues are experimentally addressed and discussed to explain why the B19V loads are significantly higher in the collagenase digested samples.

As reported by Medina et al. (ref 15 in the manuscript), plasma cells are present in the cell suspensions obtained by both methods. However, the plasma cells released by collagenase digestion have different adhesion molecules/effector profiles as well as higher maturity and survival as compared to those recovered by mechanical homogenization. Phenotypically, the B cells obtained after collagenase digestion, are resident (long-lived) memory and plasma cells, which are located in the follicular and para-follicular areas of the tonsil, while the B cells obtained by homogenization alone, correspond among others to germinal center B cells, of lower maturity, and prone to apoptosis and migration.

This we now state more clearly in the text (page 4).

It is interesting that the two methods give similar results for EBV (used here as a comparator). Given that EBV selectively infects B lymphocytes, these data would imply that B19V DNA cannot be *exclusively present in B cells*.

While B cells are the main cell accounting for EBV latency, its DNA has been found also in T cells (e.g. Assadian et al., PLoS One. 2016 May 2;11(5):e0154814), monocytes (e.g. Savard et al., J. Virol. March 2000 vol. 74 no. 6 2612-2619) as well as other cell types such as dendritic cells and epithelial cells. Our findings are in line with this.

We do not claim that B19V DNA persists exclusively in tonsillar B cells but instead, our data suggest that it is found at significantly higher copy numbers in the B-cell-enriched fraction as compared to T cells and monocytes.

Although both B19V and EBV share persistence in B cells, the life cycles are very different. For instance, EBV regularly reactivates, with constant de-novo infection of several susceptible cell populations, while B19V neither reactivates nor chronically replicates among constitutionally healthy individuals. Even if B19V would be hosted by diverse B cell subpopulations in acute (primary) infection, which is likely, according to the present data it persists for life only in the B cells with long term survival, normally present in primary and secondary lymphoid organs.

We thank the reviewer for this remark. This has now been clarified in the text (page 9).

Finally, the authors should be aware that certain tonsillar B cell subsets, such as germinal centre centrocytes and centroblasts, are highly prone to apoptosis during isolation and this may also impact upon the results.

We are grateful to the Reviewer for bringing this out, it is indeed a worthy point.

In order to minimize apoptosis, all our tissues were collected, processed and fractionated immediately after extraction. The tonsils and the derived cellular suspensions were kept on ice through all steps as recommended by Krag Kjelsen et al. (Am J Clin Pathol 2011;136:960-969). The magnetic bead separation was performed at 4°C as recommended by the manufacturer.

2. The present data do not definitively demonstrate that B19V DNA is present in B cells, as the extremely low virus loads (Figure 1c) could be entirely explained by the presence of B19V DNA in the 3-6% contamination with other cell types (line 86). Indeed the observation that EBV DNA was detectable not only in B cells, but also in the isolated T cells and monocytes (Figure 1d), strongly argues that cell contamination is a concern, thereby confusing the interpretation of the data.

As mentioned above, EBV DNA has also been reported in T cells as well as in monocytes, hence our EBV data do not imply contamination in these fractions.

Throughout the fractionation procedures, the higher B19V DNA copy numbers consistently correlated with the respective B cell population.

Furthermore, since cell purity is mainly affected by (i) clusters of cells and (ii) false-positive cell sorting (e.g. by non-specific antibody labelling of dead cells), the following measures/procedures were taken/examined:

- a) *Treatment with either DNase, trypsin or Triton X of the cell suspensions, to minimize the re-establishment of cell to cell contacts, showed neither additional benefit on cell purity nor on the quantities of B19V DNA detected (suggesting that the viral DNA indeed was intracellular).*

- b) *The forward and side scatters of the purified B and T cells in flow cytometry are compatible with a single cell profile.*
- c) *All tonsils were kept on ice and processed immediately after removal to minimize cell death.*
- d) *All tonsils were processed in exactly the same manner, whereby any contamination must be randomly reflected in all fractions.*

To convincingly demonstrate the presence of B19V DNA in B cells, the authors must use a direct approach, such as cell surface staining for CD19 combined with FISH for B19V DNA followed by FACS analysis.

We agree that such data in principle would further strengthen our findings. However, the complex experimentation would be prone to fail in this particular instance, due to the very low copy numbers of B19V DNA, contained in a low proportion of cells. The only protocol available (Manaresi et al., Journal of Virological Methods 223(2015) 50-54) does not appear to reach the high sensitivity of the qPCR used here in this work.

3. Have the authors looked for B19V DNA in circulating B cells? If present, buffy coats (pre-screened for B19V DNA positivity) could be sorted to provide high numbers of different B cell subsets and then analysed for B19V DNA. This would help address the question of which B cell types carry B19V DNA, which was hampered in the current work by the low cell yields.

Appreciating the Reviewer's suggestion, we now determined by the corresponding qPCRs, B19V and EBV DNAs in (i) peripheral blood mononuclear cells and (ii) enriched CD19+ cells from venous blood of seven B19 seropositive individuals and one seronegative control. All these cell preparations were negative for B19V DNA while positive for EBV DNA in three individuals. This has been added to the manuscript (pages 6,10,13)

The negative B19V results in peripheral blood (cf. tissue) may be due to:

- a. *The constant turnover of circulating B cells (half-life ranging from a few days to 5-6 weeks; Fulcher and Basten, Immunology and Cell Biology (1997) 75, 446–455), as opposed to resident- memory B cells that have been estimated to last for decades (Yu et al, Nature (2008); 455:532-536).*
- b. *The different phenotype and effector functions of peripheral blood B cells as compared to tonsillar B cells (Perez et al., Immunol Cell Biol. 2014 Nov;92(10):882-7 ; Medina et al., Blood. 2002 Mar 15;99(6):2154-61). Moreover, our experimental data show that circulating B cells are negative for globoside, which may have an impact on virus entry.*

4. If B cells are the reservoir for B19V, the authors should look for B19V DNA in patients receiving anti-CD20/Rituximab therapy.

Rituximab treatment has been shown to induce rapid and prolonged depletion of peripheral-blood B cells but not of B cells in lymphoid organs (Kamburova et al, Journal of Transplantation 2013; 13: 1503–1511). Indeed, immunoglobulin levels are little affected by the drug (Edwards et al. N Engl J Med 2004; 350:2572-258). As described in the previous point, we found that circulating B cells, long after primary infection, are negative for B19V DNA.

5. In the experiments described in Figure 2, can the authors comment on the proportion of cells which became VLP positive.

Based on flow cytometry, on average 8 % of the Raji cells and 3,6% of the GM 12878 cells became VLP positive.

REVIEWERS' COMMENTS:

Reviewer #1 (Remarks to the Author):

The points I raised in my initial (first round) review of the manuscript "Longevity of B cells revealed via an extinct type of human parvovirus B19" by Lari Pyöriä et al. have been addressed to my satisfaction by additional experiments (resulting in the addition of new figures and tables) and the corresponding changes in the manuscript text.

Reviewer #2 (Remarks to the Author):

In the revised version of the manuscript, the authors have made heroic efforts to address the reviewers' concerns to various levels of satisfaction. However, one major issue remains, concerning the inability to determine the phenotype of the cell in which B19 persists. For example, CD19 has been found on follicular dendritic cells, in addition to B cells. As has been pointed out by other reviewers, there are many questions about possible contamination of other cellular types that appear to be very difficult to answer. Although the lymphoid persistence of B19 is convincing, to pin it to memory B cells, as intriguing as it is, compounded with reliance on PCR assays, is problematic.

Reviewer #3 (Remarks to the Author):

This revised manuscript by Pyöriä and colleagues presents new evidence supporting the view that human B cells are a potential site of persistence of human parvovirus B19. The authors' conclusions are largely based on viral load measurements in separated B, T and monocyte/macrophage populations isolated from tonsils, with the highest amounts of B19V DNA being detected in the B cell fraction. However, with the exception of two tonsil samples with atypically high B19V loads, the authors were unable to determine whether B19V was selectively localised in particular naïve or memory B cell subsets. Similarly, attempts to identify B19V DNA in circulating B lymphocytes were unsuccessful. In conclusion, the authors have demonstrated for the first time that B cells represent a site of B19V persistence in vivo, although other possible reservoirs such as monocytes and macrophages cannot be excluded.

The authors have now satisfactorily answered my original technical concerns about some aspects of the work and clarified most of my general queries.

REVIEWERS' COMMENTS:

Reviewer #1 (Remarks to the Author):

The points I raised in my initial (first round) review of the manuscript “Longevity of B cells revealed via an extinct type of human parvovirus B19” by Lari Pyöriä et al. have been addressed to my satisfaction by additional experiments (resulting in the addition of new figures and tables) and the corresponding changes in the manuscript text.

We are pleased that we could address the points to the satisfaction of the Reviewer

Reviewer #2 (Remarks to the Author):

In the revised version of the manuscript, the authors have made heroic efforts to address the reviewers' concerns to various levels of satisfaction. However, one major issue remains, concerning the inability to determine the phenotype of the cell in which B19 persists. For example, CD19 has been found on follicular dendritic cells, in addition to B cells. As has been pointed out by other reviewers, there are many questions about possible contamination of other cellular types that appear to be very difficult to answer. Although the lymphoid persistence of B19 is convincing, to pin it to memory B cells, as intriguing as it is, compounded with reliance on PCR assays, is problematic.

We agree with the Reviewer that determining the exact subpopulation of B cells where B19V persists would have been preferable. This is touched upon in the discussion session of the manuscript.

The possibility of contamination brought up by Reviewer 3 in the past, we've already ruled out in our previous rebuttal letter, to his/her satisfaction (see below).

We trust that the sample size, statistical significance, technical precautions during the sample processing as well as the proven susceptibility of B cells to uptake B19V, together justify our conclusions.

Reviewer #3 (Remarks to the Author):

This revised manuscript by Pyöriä and colleagues presents new evidence supporting the view that human B cells are a potential site of persistence of human parvovirus B19. The authors' conclusions are largely based on viral load measurements in separated B, T and monocyte/macrophage populations isolated from tonsils, with the highest amounts of B19V DNA being detected in the B cell fraction. However, with the exception of two tonsil samples with atypically high B19V loads, the authors were unable to determine whether B19V was selectively localised in particular naïve or memory B cell subsets. Similarly, attempts to identify B19V DNA in circulating B lymphocytes were unsuccessful. In conclusion, the authors have demonstrated for the first time that B cells represent a site of B19V persistence in vivo, although other possible reservoirs such as monocytes and macrophages cannot be excluded.

The authors have now satisfactorily answered my original technical concerns about some aspects of the work and clarified most of my general queries.

We are pleased that we could address the points raised by the Reviewer to his/her satisfaction.